# Accessing Mole-Scanning through Community Pharmacy: A Pilot Service in Collaboration with Dermatology Specialists

**DOI:** 10.3390/pharmacy8040231

**Published:** 2020-12-03

**Authors:** Charlotte L. Kirkdale, Zoe Archer, Tracey Thornley, David Wright, Mette Valeur, Nicola Gourlay, Kurt Ayerst

**Affiliations:** 1Boots UK, Thane Road, Nottingham NG90 1BS, UK; tracey.thornley@boots.co.uk (T.T.); Nicola.Gourlay@boots.co.uk (N.G.); 2ScreenCancer UK Ltd., Innovation Centre, Maidstone Road, Chatham, Kent ME5 9FD, UK; zoe.archer@screencancer.com (Z.A.); mette.valeur@screencancer.com (M.V.); kdayerst@usa.net (K.A.); 3School of Pharmacy, University of Nottingham, University Park, Nottingham NG7 2RD, UK; 4School of Pharmacy, University of East Anglia, Norwich Research Park, Norwich NR4 7TJ, UK; d.j.wright@uea.ac.uk

**Keywords:** mole, pharmacy, melanoma, scanning, dermatoscopy

## Abstract

Early identification and treatment of malignant melanoma is crucial to prevent mortality. The aim of this work was to describe the uptake, profile of users and service outcomes of a mole scanning service in the community pharmacy setting in the UK. In addition, health care costs saved from the perspective of general practice were estimated. The service allowed patients to have concerning skin lesions scanned with a dermatoscopy device which were analyzed remotely by clinical dermatology specialists in order to provide recommendations for the patient. Patients were followed up to ascertain the clinical outcome. Data were analyzed for 6355 patients and 9881 scans across 50 community pharmacies. The majority of the scans required no further follow-up (n = 8763, 88.7%). Diagnosis was confirmed for 70.4% (n = 757/1118) of scans where patients were recommended to seek further medical attention. Of these, 44.3% were ultimately defined as normal (n = 335) and 6.2% as malignant melanoma (n = 47/757). An estimated 0.7% of scans taken as part of the service led to a confirmed diagnosis of malignant melanoma. This service evaluation has shown that a mole scanning service available within community pharmacies is effective at triaging patients and ultimately playing a part in identifying diagnoses of malignant melanoma.

## 1. Introduction

In 2017, the World Health Assembly passed a resolution urging governments and the World Health Organisation to scale up national cancer prevention and control in order to reduce premature mortality from cancer [1]. This will require greater awareness of symptoms of cancer and accelerated access to diagnosis and treatment; both of particular importance in the case of melanoma skin cancer. The fifth most common cancer in the United Kingdom (UK), melanoma skin cancer, accounts for 4% of new cancer cases and 1% of cancer deaths [2] and its incidence is still increasing (135% since the 1990s [2]).

Although recommendations to screen higher-risk individuals are made in some countries (Australia, New Zealand, Germany, the Netherlands and the UK [3]) national melanoma screening programs are not widely recommended (Germany, and pilots in Australia and the US [4,5]); and a recent Cochrane review (2019) determined that general adult screening of melanoma was not refuted by evidence [6]. Instead, the UK pursues education programs to raise awareness of signs and symptoms, risk factors and the need to self-check to promote earlier detection and thus reduce mortality. Early diagnosis and treatment of malignant melanoma is key not only in improving outcomes for the patient but also from an economic perspective [7,8]. Kakushadze et al. observed a more than three-fold increase in melanoma treatment costs between stages II and III [9].

Current practice in the UK is for a patient to visit their primary care practitioner (or General Practitioner (GP)) if they are worried about a suspicious mole or lesion. The GP will inspect and measure the mole and refer using the National Institute for Health and Care Excellence (NICE) suspected cancer pathway referral for melanoma seven point checklist [10]. GPs deal with 13 million consultations for skin disorders a year and 880,000 of these cases are referred to secondary care (hospitals) [11]. GPs may undergo only short periods of dermatology specialization during training, and in some areas this is not compulsory [12], yet they are the ‘gatekeeper’ and responsible for the initial assessment and referral to specialist care for those seeking help through the National Health Service (NHS) [13]. One systematic review found that referral accuracy of GPs ranged from 0.70 to 0.88 for sensitivity and 0.70 to 0.87 for specificity [14], suggesting that there is more that can be done to improve the appropriateness of referrals.

Outside the NHS, different models are available for the identification of suspicious moles, from traditional dermatology clinics (healthcare professional inspects mole) to mobile phone Apps (patients phone camera used to take photos and keep track of moles and send reminders to repeat on a regular basis) and teledermatoscopy (dermascopic images of moles and lesions taken and sent to be remotely interpreted by experts). The use of dermoscopy is fundamental to ensure accurate interpretation and referral. Dermoscopy is a non-invasive scanning technique, which is more accurate than naked-eye examination [15] and highlights features which can indicate pathological changes consistent with melanoma [16,17]. Use of these types of devices by GPs has been found to have significantly improved sensitivity as compared with naked-eye examination alone [16]. Use of dermoscopy has also been found to improve the probability of diagnosis by 1.25 times and be cost effective compared to without (incremental cost-effectiveness ratio of EUR 89 (95% CI−EUR 60 to EUR 598)) [17].

Community pharmacies are positioned in easily accessible locations [18]. Services are often available without the need for an appointment and are also offered outside of usual working hours, such as evenings and weekends. Community pharmacy offers both private (for example; flu, chickenpox and travel vaccinations, test and treat services) and NHS services [19], and are increasingly aiming to help reduce the burden on other areas of the health service. The potential of community pharmacy teams has been recognized in the NHS Interim People Plan; the expectation being that the public will have access to an increasing range of clinical care provided by pharmacy professionals [20].

A private mole scanning service in the community pharmacy setting has also been evaluated in Norway [21]. Pharmacies across Boots Norge AS set up a mole scanning service in 2010 in cooperation with ScreenCancer and using dermoscopy. An evaluation of the service found the approach was acceptable to patients and in 2014 the service identified 4.1% of the melanoma cases registered in the Norwegian Cancer Registry [21].

This service evaluation aimed to describe the uptake of a mole scanning service in the community pharmacy setting in the UK, to describe the population accessing the service and service outcomes, and to estimate health care costs saved from the perspective of general practice.

## 2. Materials and Methods

### 2.1. Study Overview

The project was reviewed by the Boots research governance board and deemed a service evaluation, using anonymized data. Ethical approval was therefore not required.

### 2.2. Intervention

Pharmacists and their teams carried out the initial patient facing consultation for the mole scanning service which was delivered in collaboration with ScreenCancer who were responsible for the operation of the service. Pharmacy team members providing the service were required to read both a service and clinical guide, which detailed how to deliver the consultation and scan safely as well as how to educate the patients on how to check their own moles using the ABCDE criteria (asymmetry, border irregularity, color that is not uniform, diameter >6 mm, evolving size, shape or color) [22]. The training also included watching educational videos and conducting 3 practice scans on colleagues to submit to trained health advisors or registered nurses for review before being authorized to provide the service to the public. In addition, pharmacy staff also had to have a current Enhanced Disclosure Barring Service Check (DBS Check) because they were taking photos of patients.

Patients may have seen marketing materials for the service, booked an appointment online, referred by their GP (pharmacy teams notified local GP practices of the service during set up) or made aware of the service during a conversation with the pharmacy team. The pharmacy team then explained the service, eligibility criteria, chaperone policy and costs to the patient using a service leaflet. The service was not available for moles in intimate areas of the body. Any patients presenting with a concern but who were ineligible to continue were signposted to their GP. The cost to the patient of the initial mole scanned was GBP 35, plus GBP 15 for each additional mole, with a suggestion that the patient should consider seeing their GP if more than 4 or 5 moles were concerning them.

Once the patient had agreed to the service and were deemed eligible, the discussion moved to the consultation room where further details were recorded on the ScreenCancer Mole Navigator^®^ software. Patient consent was taken and the chosen mole(s) were scanned, their mole history taken (using the ABCDE rule) [22] and images of the mole(s) captured. Digital dermatoscopes were used to take clinical images (15 cm distance from the mole) to give an overview of the condition of the skin, and dermatoscopic images taken at close range using a nose cone attachment to show color, shape and intensity of the area under scrutiny. (SIAscope [23] from February 2016 to April 2019 and the ScreenCancer Optical Transfer Diagnosis (OTD) dermatoscopic camera (Balter Medical^TM^, Bergen, Norway) from April 2019). Before completing the consultation, patients were provided with a copy of the customer record form (containing a unique 6-digit code which linked to their results) and sun care advice. The patient data and images were accessed remotely by dermatology specialists who analyzed the images, patient history and questionnaire results. Personal reports were sent to patients within 1 week. Where analysis suggested the mole or lesion to be at risk (suspicious or possible Basal Cell Carcinoma), attempts were made by a Dermatology Nurse or Healthcare Advisor to contact patients to discuss next steps, with a subsequent follow up call after 3–6 months.

The service was initiated in February 2016 across 50 community pharmacies (44 in England, 3 Scotland, 2 Northern Ireland, 1 Wales) that were part of a national pharmacy chain. The community pharmacies providing the service were predominantly larger, with some representation from smaller local community pharmacies.

### 2.3. Data Collection and Analysis

Data used in the service evaluation were collected through the Mole Navigator® system (date, time and location of consultation, age, gender, post code, outcome of recommendation from service, outcome of follow-up call). Data were aggregated and anonymized before being analyzed in Excel 2013 using descriptive statistics. For the purposes of describing socioeconomic status of those accessing the service, deprivation profiles were calculated using the Carstairs index [24]. This is based on 4 census indicators: low social class, lack of car ownership, overcrowding and male unemployment. A negative value indicates areas of low deprivation and a positive value equates to high deprivation.

Estimation of costs to the NHS were based on delivery of the service by 5500 pharmacies and 52 consultations per year per pharmacy (average seen in trial), a referral rate of 0.113 and cost of GP consultation of GBP 39.23 [25].

## 3. Results

Between 12 February 2016 and 31 December 2019, 6110 patients accessed the service for a total of 6354 consultations across 50 community pharmacies (5431 consultations in England, 522 in Scotland, 202 in Northern Ireland and 199 in Wales). Patients mean age was 50 years (range 12–95 years (8 unknown)), median 45 years, mode 33 years and the majority of patients were female (n = 4416, 72.3%; Figure 1). The deprivation spread of patients was similar when compared to the profile of general retail healthcare customers visiting the same pharmacies over the study period, with a skew towards more affluent patients accessing the private service (Figure 2). Consultations peaked in the summer months (June–September) and were equally spread throughout weekdays (17.0–17.5% per day), with fewer consultations at the weekends (10.8% on Saturdays and only 2.6% on Sundays). Consultation times also peaked around lunchtime (40.9% between 11 am and 1 pm).

Patients were able to receive multiple consultations and have multiple moles scanned and during the evaluation period 9880 scans were made; the average number of moles checked per patient was 1.62 (range 1–17). A small number of patients attended more than one consultation (3.5%, n = 216), with the average number of consultations for these patients being 2.1 (range 2–5).

The majority of the scans required no further follow-up (n = 8762, 88.7%); with 57.4% deemed normal, 22.4% non-suspicious keratosis and 6.1% potential sun damage (see Table 1). Dermatology specialists recommended follow up for 11.3% (n = 1118) of scans due to suspicious lesion (3.0%), other skin lesions (3.0%), sun damage (2.0%), inability to exclude basal cell carcinoma (1.8%) or further information required (four scans; see Table 1).

Attempts were made to contact all patients who required follow-up, with diagnosis confirmed for 70.4% (n = 757/1118) of these scans (the rest were due to inability to make contact or patient deciding not to share the information). Results of this follow up can be seen in Table 2. Malignant melanoma was diagnosed in 6.2% of lesions (n = 47/757). There were no patients who received multiple melanoma diagnoses. If the same percentage was assumed to be correct for the rest of the remaining unconfirmed lesions (n = 361), an extrapolated additional 22 scans would have had a confirmed diagnosis of malignant melanoma, meaning that overall 0.7% (n = 69/9880) of scans taken as part of the service are likely to have led to a confirmed diagnosis of malignant melanoma.

In England in 2017 there were 13,740 registered newly diagnosed cases of malignant melanoma of the skin [26]. If the service had not been available, it could have resulted in patients accessing their GP unnecessarily, at a cost of GBP 344,000 (based on 8762 patients not requiring further investigation). Extending the service nationally could deliver annual savings of up to GBP 10 m, not taking into account the additional benefits of increasing awareness and prevention measures, and earlier diagnosis. A further GBP 1.3 m per annum could potentially be saved by allowing pharmacists to refer patients with suspicious moles directly to consultant dermatologists in secondary care using the Independent sector treatment centers model.

## 4. Discussion

These results demonstrate that more than one in ten people accessing the service required clinical follow-up, with 133 patients (one in six) receiving a formal diagnosis of cancer as a result. One quarter of referrals received confirmation of sun damage, which is still a recognized cause of melanoma [7], and a small number received other unexpected diagnoses which may improve their quality of care in the future. Nearly 90% of consultations did not require referral, therefore this would have prevented a large number of people needing to see their GP and saved the NHS resources. However, nearly half of patients’ lesions were not deemed ‘normal’ and therefore these patients still received reassurance and additional support from the community pharmacy teams, e.g., referral, advice around skin care and sun protection, and prevention and monitoring.

The high volume of consultations provided illustrates that this is a well utilized service, showing seasonal changes in demand and higher demand during weekdays and lunchtimes (likely reflecting more limited opening times on weekends). However, with an average of one consultation per pharmacy per week, this is still a manageable workload for community pharmacy teams to adopt. The majority of patients were female, which is consistent with a previous study reporting on use of community pharmacy [27], but provides valuable insights when considering marketing strategies to enhance reach and access. Skin lesions accumulate with age and melanoma incidence is highest in individuals over 55 years [28]. The age profile of patients accessing the service show that patients begin to seek advice before this age; demonstrating the potential to again support early diagnosis.

Patients represented a variety of socioeconomic backgrounds, although as expected for a private service, there were more in the less deprived categories. Whilst this private service can improve outcomes for those who can afford to access it, and as a result further exacerbate health inequalities, it could be argued that it frees up time for others to access their GP for the same purpose.

Provision of sufficient services for worried patients to have their lesions checked must be ensured. Whilst we do not know at what stage patients accessed the community pharmacy service, the accessibility and convenience will no doubt play a part in a patient’s decision whether or not to have a mole checked at an early stage. This is where community pharmacy could prove to be attractive as an option for patients who may perhaps not have had a check otherwise. In addition, they are well placed to raise awareness of the risks of skin cancer in day-to-day interactions with customers seeking skin care and sun care advice. The knowledge of these healthcare professionals can be utilized to encourage self-checking and where any concerns arise, support patients through mole scanning services (NHS and private) and support national cancer awareness campaigns.

The COVID-19 pandemic has impacted patient access to healthcare services. Not only has there been a marked reduction in A&E attendance figures [29], but also of patients with suspicious moles; one service reported a 34% reduction in referrals in February–April 2020 compared to 2019 [30]. It is therefore even more important that patients are given opportunities to have their skin lesions checked, particularly in the case of a suspected cancer where early detection is critical to survival. Provision of an alternative service through community pharmacy may be welcomed as a way to reduce pressure on GP surgeries going forward, particularly as they are likely to be playing ‘catch up’ following lockdown measures bought into place to prevent the spread of COVID-19. The pandemic has also brought the use of new technologies, such as telemedicine, to the fore, highlighting the need to adopt such practices in new services to widen accessibility.

This evaluation benefits from the use of a rigorous testing process and a large sample size (9880 scans collected across the UK). The self-paid service was provided across one pharmacy organization in collaboration with ScreenCancer which must be acknowledged as a limitation as it restricts the generalizability of the findings and may be argued to attract the ‘worried well’ and are also only applicable to the UK health setting. However, it does mean that protocols are likely to be better adhered to across sites. One quarter of the referred patients were unable or unwilling to be contacted for follow-up and there was no further follow-up of patients, and therefore, no guarantee that the lesions did not show further malicious development after this contact. However, there is also no reason to expect differing results from this cohort. The technology used for scanning was also changed from a SIAscope to the ScreenCancer OTD dermatoscopic camera. This change could have impacted detection accuracy; however, it is unlikely to have changed the outcomes of the analysis as the change occurred in 2019 and the images were still assessed by the same healthcare experts at SreenCancer. In addition, like existing pathways for melanoma detection, this service may also have missed cases of melanoma which cannot be quantified as patients with lesions initially defined as non-suspicious were not followed up.

The estimation of savings to the NHS presented here is a simplistic representation and does not take into account all costs/benefits. For example, earlier diagnosis leading to improved prognosis and time saved would also impact these calculations, therefore it is likely to be an underestimate of overall savings to the healthcare system.

## 5. Conclusions

This service evaluation has shown that a mole scanning service positioned in community pharmacy is effective at triaging patients with concerning lesions and ultimately playing a part in identifying diagnoses of malignant melanoma. Our results suggest that a mole scanning service in community pharmacy funded by the healthcare system could help to reduce pressure and additional workload on primary care and could promote earlier diagnosis and therefore result in improved prognosis for patients with malignant melanoma. A service of this nature, with good sensitivity and specificity, could also result in reduced unnecessary referrals to secondary care for specialist assessment. Further research, including comparison with existing healthcare pathways and exploration of accuracy, along with economic modelling of this service, is warranted to enable the cost-effectiveness of screening for skin cancer via this route to be estimated.

## Figures and Tables

**Figure 1 pharmacy-08-00231-f001:**
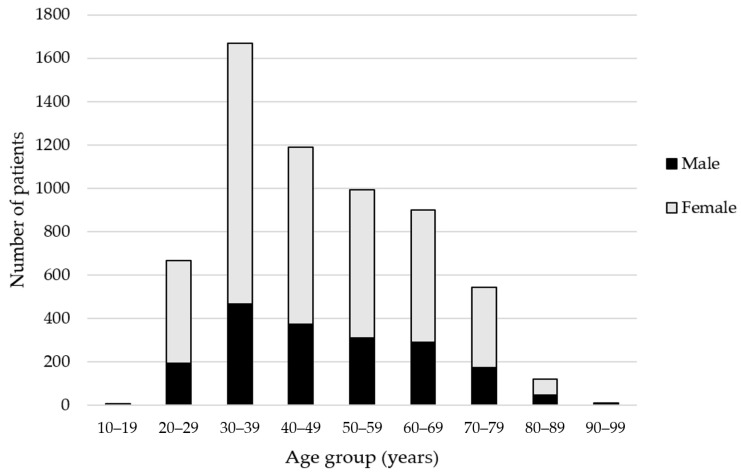
Patient age by age group in years (unknown n = 8).

**Figure 2 pharmacy-08-00231-f002:**
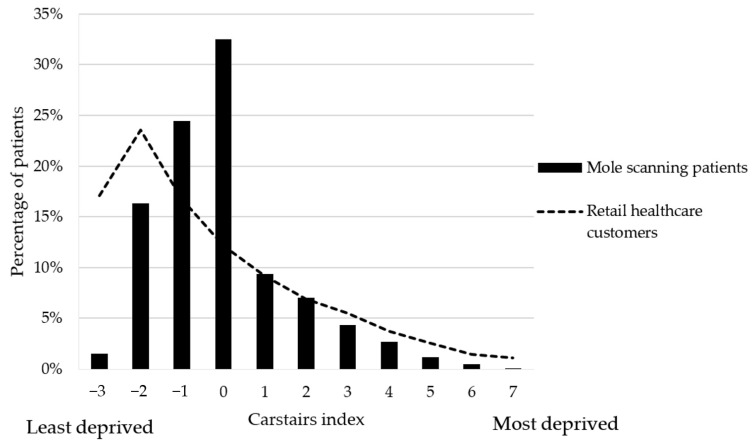
Patient and retail healthcare customer deprivation spread by Carstairs index; with −3 being the least deprived, and +7 being the most deprived (unknown for n = 108) [24].

**Table 1 pharmacy-08-00231-t001:** Outcome of ScreenCancer dermatology specialist review of scan.

	Outcome of Scan Analysis	n (% of Total)
No follow up required	Normal	5667 (57.4%)
	Non suspicious seborrhoeic keratosis	2217 (22.4%)
	Potential sun damage	603 (6.1%)
	Other skin lesion (normal)	169 (1.7%)
	Normal with atypical characteristics	106 (1.1%)
	Total–no follow up required	8762 (88.7%)
Follow up required	Suspicious	452 (4.6%)
	Other skin lesion to follow up	296 (3.0%)
	Sun damage	193 (2.0%)
	Basal cell carcinoma cannot be excluded	173 (1.8%)
	Inadequate scan/more information required	4 (0.0%)
	Total–follow up required	1118 (11.3%)
Total		9880

**Table 2 pharmacy-08-00231-t002:** Outcome of contacting patient to ascertain confirmed diagnosis of the scan/lesions (n = 757).

Confirmed Diagnosis	n (%)
Normal	335 (44.3%)
Sun damage	197 (26.0%)
Basal cell carcinoma	83 (11.0%)
Other skin condition	50 (6.6%)
Malignant melanoma	47 (6.2%)
Seborrhoeic keratosis	23 (3.0%)
Atypical	19 (2.5%)
Squamous cell carcinoma	3 (0.4%)
Total	757

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
