# Peer review of "Accessing Mole-Scanning through Community Pharmacy: A Pilot Service in Collaboration with Dermatology Specialists"

_pharmacy, 2020, doi:10.3390/pharmacy8040231_

Round 1

Reviewer 1 Report

The manuscript by Kirkdale et al describes a mole scanning service offered by community pharmacies. Overall, this is an interesting approach in melanoma prevention and the paper is well written.

A few comments:

The authors conclude by recommending to establish this service as an effective way of triaging patients with concerning lesions. The present study, however, does not give any results regarding specificity, negative predictive values and positive predictive values of the scanned moles. Therefore, the effectiveness and accuracy of this approach, also compared to the current practice via General Practitioner, can not be fully evaluated. Although the results of this pilot project look promising and the approach does have certain benefits, more research is needed before establishing this method as an alternative way of diagnosing concering lesions.

Author Response

The manuscript by Kirkdale et al describes a mole scanning service offered by community pharmacies. Overall, this is an interesting approach in melanoma prevention and the paper is well written.

A few comments:

The authors conclude by recommending to establish this service as an effective way of triaging patients with concerning lesions. The present study, however, does not give any results regarding specificity, negative predictive values and positive predictive values of the scanned moles. Therefore, the effectiveness and accuracy of this approach, also compared to the current practice via General Practitioner, can not be fully evaluated. Although the results of this pilot project look promising and the approach does have certain benefits, more research is needed before establishing this method as an alternative way of diagnosing concerning lesions.

We are unable to make any claims on specificity, negative predictive values or positive predictive values from the data we have access to. The conclusion has therefore been amended to remove the recommendation that it is established as a triage method and to be clear that further research is needed.

Line 253-260 amended to now read: “Our results suggest that a mole scanning service in community pharmacy funded by the healthcare system could help to reduce pressure and additional workload on primary care and could promote earlier diagnosis and therefore result in improved prognosis for patients with malignant melanoma. A service of this nature, with good sensitivity and specificity, could also result in reduced unnecessary referrals to secondary care for specialist assessment. Further research, including comparison with existing healthcare pathways and exploration of accuracy, along with economic modelling of this service is warranted to enable the cost-effectiveness of screening for skin cancer via this route to be estimated.”

Reviewer 2 Report

The authors provide the medical results of mole scanning through a community pharmacy service. They also try to evaluate the costs and financial benefits to the UK health care system.

I have the following suggestions:

Materials and Methods

1) It would be interesting to know how and to whom the 35 pounds per initial scanning were distributed and who payed how much for the scanning equipment. This is an important information to assess the financial benefits.

Results

2) The authors should supply more data (tables or figures) on sex and age distribution.

Discussion

3) There has been a change in the used technology from Siascope to OTD. It should be mentioned in the discussion that this change could have an impact on detection accuracy, but will probably not change the outcome of the analysis as the change took place only in 2019.

4) It is not clear whether 47 melanomas were detected in 6110 patients or whether some patients had multiple melanomas.

5) The detection rate of approx. 0.7% should be compared to the assumed melanoma prevalence in the UK population and the screened age group (if these data are available) and to the detection rate of screening programs in oher countried, for example Germany and Australia.

6) One might speculate on the melanomas which might have been missed by this approach.

7) It should be discussed that these findings are specific for the UK health system.

Author Response

The authors provide the medical results of mole scanning through a community pharmacy service. They also try to evaluate the costs and financial benefits to the UK health care system.

I have the following suggestions:

Materials and Methods

1) It would be interesting to know how and to whom the 35 pounds per initial scanning were distributed and who payed how much for the scanning equipment. This is an important information to assess the financial benefits.

This was a private service which the patient paid to have access to as explained in Line 108-109: “The cost to the patient of the initial mole scanned was £35, plus £15 for each additional mole”. The payment was made to the pharmacy. The service is jointly run by the pharmacy chain (Boots) and ScreenCancer but the specific commercial nature is sensitive and unfortunately cannot be shared.

Results

2) The authors should supply more data (tables or figures) on sex and age distribution.

We have added in a figure showing spread of patients by age and sex (Figure 1 Line153).

Discussion

3) There has been a change in the used technology from Siascope to OTD. It should be mentioned in the discussion that this change could have an impact on detection accuracy, but will probably not change the outcome of the analysis as the change took place only in 2019.

Added the following text to Line 239 to account for this: “The technology used for scanning was also changed from a SIAscope to the ScreenCancer OTD dermatoscopic camera. This change could have impacted detection accuracy; however is unlikely to have changed the outcomes of the analysis as the change occurred in 2019 and the images were still assessed by the same healthcare experts at SreenCancer.”

4) It is not clear whether 47 melanomas were detected in 6110 patients or whether some patients had multiple melanomas.

The figure of 47 melanomas in 6110 patients also relates to 47 patients as there were no patients with multiple melanomas diagnosed. It would also be unlikely for a patient to present with two separate melanomas at different times as most patients that have been diagnosed with a melanoma would remain under hospital review up to 5 years depending on the depth of the melanoma. Line 172 has been updated to more clearly express this. “There were no patients who received multiple melanoma diagnoses.”

5) The detection rate of approx. 0.7% should be compared to the assumed melanoma prevalence in the UK population and the screened age group (if these data are available) and to the detection rate of screening programs in other countries, for example Germany and Australia.

We are unaware of known detection rates for the UK population to compare ours to. We were able to add in the number of cases of melanoma in the England (see line 179: “In England in 2017 there were 13,740 registered newly diagnosed cases of malignant melanoma of the skin[26].”). As this was not a screening programme, and the results of this evaluation are only applicable to the UK healthcare system, we did not feel it beneficial to make comparisons with other countries.

6) One might speculate on the melanomas which might have been missed by this approach.

Line 243 added: “In addition, like existing pathways for melanoma detection, this service may also have missed cases of melanoma which cannot be quantified as patients with lesions initially defined as non-suspicious were not followed up.”

7) It should be discussed that these findings are specific for the UK health system

Line 234-235 have been amended to read : ”The self-paid service was provided across one pharmacy organization in collaboration with ScreenCancer which must be acknowledged as a limitation as it restricts the generalizability of the findings and may be argued to attract the ‘worried well’ and are also only applicable to the UK health setting.”

Round 2

Reviewer 2 Report

The authors responded to the comments of the reviewer.